

# A scanning quantum cryogenic atom microscope at 6 K

Stephen F. Taylor[1,2], Fan Yang[1,2], Brandon A. Freudenstein[2,3] and Benjamin L. Lev[1,2,3⋆]

**1** Department of Applied Physics, Stanford University, Stanford, CA 94305, USA
**2** E. L. Ginzton Laboratory, Stanford University, Stanford, CA 94305, USA
**3** Department of Physics, Stanford University, Stanford, CA 94305, USA

⋆ benlev@stanford.edu

## Abstract

The Scanning Quantum Cryogenic Atom Microscope (SQCRAMscope) is a quantum sensor in which a quasi-1D quantum gas images electromagnetic fields emitted from a nearby sample. We report improvements to the microscope. Cryogen usage is reduced by replacing the liquid cryostat with a closed-cycle system and modified cold finger, and cryogenic cooling is enhanced by adding a radiation shield. The minimum accessible sample temperature is reduced from 35 K to 5.7 K while maintaining low sample vibrations. A new sample mount is easier to exchange, and quantum gas preparation is streamlined.

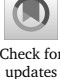

# 1   Introduction

The Scanning Quantum Cryogenic Atom Microscope (SQCRAMscope) is a scanning probe quantum sensor that combines cryogenics with the techniques of laser cooling and trapping of neutral atoms [1,2]. It utilizes an ultracold Bose-Einstein condensate (BEC) of atoms to sense fields. The atoms are confined in a 1D cigar-shaped trap by a highly anisotropic magnetic field produced by an "atom chip" trapping device [3]. The atoms can be positioned within a micron of the sample's surface and scanned anywhere within an approximately $3 \times 5$ mm$^2$ area. Magnetic, electric, and Casimir-Polder potentials may be imaged while sample temperatures are tuned from room to cryogenic temperatures. Through modulations of density or spin state, the atoms transduce electromagnetic potentials generated by the sample into a signal that can be imaged on a CCD camera. In a single shot, a 1D measurement of the potential is made along the length of the BEC. The atoms can further be scanned over the material surface, allowing 2D field images to be recorded. In the context of magnetometry, the source current or magnetization pattern may be determined by measuring the atom–to–surface distance and inverting the Biot-Savart law.

We have recently demonstrated the SQCRAMscope's scientific utility by imaging electron nematic transport in an iron-based pnictide superconductor via magnetometry [4]. The SQCRAMscope is among the best micron-scale low-frequency magnetic field sensors available [2], can detect a quarter of an electric charge a few microns away [5], and is capable of sensing the Casimir-Polder force from a conductor within a micron of the atoms.

Cryogenic cooling of the sample enables the exploration of exotic phase transitions and other temperature-dependent phenomena. However, combining atomic cooling and trapping techniques with the cryogenic cooling of nearby materials is technically challenging. Such systems have been built for the purpose of, e.g., trapping atoms with superconducting wires and/or near superconducting materials [6–15], increasing the lifetime of trapped ions [16, 17], and creating hybrid quantum information devices by coupling atoms to superconducting qubits [18]. Scanning probe sensing with a BEC or ultracold thermal gas has been previously demonstrated [5, 19–22]; however, no other apparatus but the SQCRAMscope serves as a scanning probe for quantum materials with the capability of rapid sample exchange and BEC recovery, high-numerical-aperture imaging, and wide-area sample imaging [1,2].

An earlier version of the SQCRAMscope was limited to sample temperatures above 35 K. Unfortunately, this limited its ability to explore lower-temperature many-body phenomena, such as the unconventional superconductivity that sets in below the nematic transition in the iron-based pnictide Ba(Fe$_{1-x}$Co$_x$)$_2$As$_2$ [23,24]. Moreover, the microscope used a liquid helium flow cryostat, rather than a more efficient closed-cycle system, and was therefore expensive to operate. Lastly, the previous SQCRAMscope version employed both a cumbersome sample mounting scheme and an inefficient quantum gas preparation procedure [1].

This work presents the results of a redesign of the SQCRAMscope to include a radiation shield and a closed-cycle cryostat without adding deleterious vibrations. A new sample mount design accommodates the radiation shield while also providing an easier method for sample exchange. The newly incorporated closed-cycle cryocooler enables continuous operation of the cryostat without the need to replenish cryogens. It is specially designed to limit the transmission of vibrations to the sample. The custom radiation shield reduces the radiative heat load on the sample stage while accommodating all necessary optical access and a large range of motion of the sample. Together, these modifications enable operation down to 5.7 K, a sixfold improvement in temperature. Accessing lower temperatures allows the SQCRAMscope to now explore a far wider array of phenomena exhibited by quantum materials.

We describe the new aspects of the SQCRAMscope hardware in Sec. 2, including the new cryogenic cooling system and radiation shield. The new atom chip and ultracold atom produc-

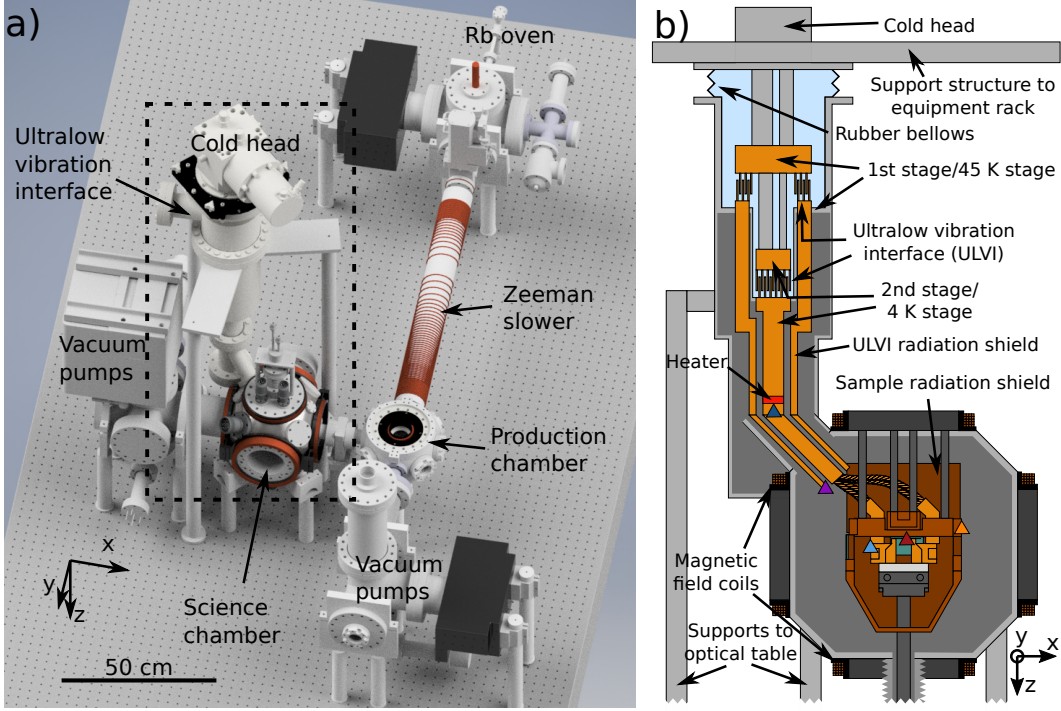

Figure 1: Schematic of the SQCRAMscope. (a) Rendering of the SQCRAMscope as mounted on the optical table. An equipment rack suspended from the ceiling (not shown) secures the cold head above the optical table. (b) Cross-section schematic of the inside of the science chamber (not to scale); region corresponds to the dashed black box in panel (a). Five temperature sensors are labeled by colored triangles: silicon diodes on the ULVI second stage (dark blue), ULVI radiation shield (purple), and sample mount (light blue), and a calibrated silicon diode on the sample chip (red) as well as a platinum sensor on the radiation shield (orange).

tion procedure are summarized in Sec. 3. The base temperature and vibrational properties of the improved system are presented in Sec. 4.

## 2  Apparatus improvements

All measurements take place in an ultrahigh vacuum (UHV) chamber designed for the production of ultracold atomic gases, as shown in Fig. 1. Rubidium-87 atoms from an oven are initially trapped and cooled in the production chamber before being transported by an optical tweezer to the science chamber where measurements are performed. A sample is attached to the bottom of a thin silicon chip within the science chamber, where it can be scanned relative to the atoms by a translation stage; see Refs. [1,2] for details. A closed-cycle cryostat, radiation shield, and heater allow the sample temperature to be tuned from room temperature down to cryogenic temperatures. More detailed descriptions of the cryostat, sample mount, and radiation shield are provided in the following subsections.

### 2.1  Cryogenics

The previous version of the SQCRAMscope utilized a liquid-flow cryostat to cool the sample. While adequate in terms of vibration, this wet cryostat has the significant disadvantage of

consuming a prodigious amount of liquid helium, a resource of increasing scarcity. Repeatedly replenishing it while running is both costly and time-consuming, hindering continuous operation of the system at low temperature.

To lessen the need for liquid cryogens, the cryogenic refrigerator of the SQCRAMscope has been replaced with a two-stage, closed-cycle pulse tube cryocooler (Sumitomo SRP-082B2S). This provides a nominal cooling power of 0.9 W at the second stage when operated at 4.2 K. The primary downside to switching to a closed-cycle system is the increased vibrations arising from moving parts within the cold head. To mitigate this problem, all moving parts are confined to a remote valve unit that is separated from the cold head. Additionally, the compressor driving the cold head (Sumitomo F-70LP) is located in a separate room to reduce vibrations present at the optical table. Finally, a custom ultralow vibration interface (ULVI) for the cold head limits vibration transmission to the sample by physically isolating it from the UHV chamber. (The ULVI was designed by and sourced from ColdEdge Technologies.) This isolation is accomplished by interlacing copper heat-exchanger fins mounted to the cold head with matching fins on the vacuum chamber side without physical contact; see the schematic in Fig. 1b. A low-pressure helium gas surrounding the fins transmits cooling power to the vacuum chamber side without directly coupling vibrations.

The cold head is mounted to an equipment rack suspended from the ceiling, and so it is connected to the optical table and the experimental UHV chamber by only a rubber bellows that seals the helium exchange gas within the ULVI. The temperature of the second stage within the ULVI is monitored using a silicon diode temperature sensor, and a nearby heater controls the cryostat temperature. (The heater is shown in red in Fig. 1b.) The 4 K cold-finger of the ULVI enters the main vacuum chamber at a diagonal and attaches to the sample mount with two copper braids. The flexible braids allow the sample mount to move while heat-sunk to the cold finger.

Within the ULVI, the second stage of the cryostat is shielded from room-temperature blackbody radiation by a gold-plated copper shield cooled by the first stage of the cryostat; see schematic in Fig. 1b. Once within the science chamber, this shield overlaps with, but does not touch, a separate radiation shield connected to the atom chip mount. This second shield is cooled separately by a liquid nitrogen reservoir; see description in Sec. 2.3.

## 2.2 Sample mount

The sample mount, shown in Fig. 2, is designed for a large range of motion and easy sample loading while maintaining effective cooling. To minimize the distance between the atom chip and the sample, which directly impacts the spatial resolution and sensitivity of the SQCRAMscope, the sample is mounted on the underside of a 150-$\mu$m thin silicon chip. This chip is secured with indium solder to two copper blocks that are cooled by copper braids attached to the 4 K stage of the ULVI. Indium solder is used to secure the silicon chip because of its high thermal conductivity and mechanical strength. A Macor ceramic spacer compensates for differential thermal contraction between the silicon chip and the rest of the sample mount. This minimizes the strain put on the silicon chip during cooling and prevents buckling or shattering due to thermal contraction. A stainless steel base attaches to the Macor's bottom surface and completes the sample mount, as shown in Fig. 2a. Two slots in the stainless steel mate with tabs on a complementing piece on the support tube to allow for easy insertion and removal; see top of Fig. 2b.

The support tube is a thin-walled Grade 9 titanium tube welded to a blank vacuum flange. The flange is connected to the vacuum chamber by a flexible bellows, allowing sample movement while under vacuum. The material and shape of the titanium tube maintain mechanical rigidity of the sample support while also minimizing thermal conduction from the cold sample to the room-temperature vacuum flange. A hexapod 6-axis positioning system (PI H-811.D2)

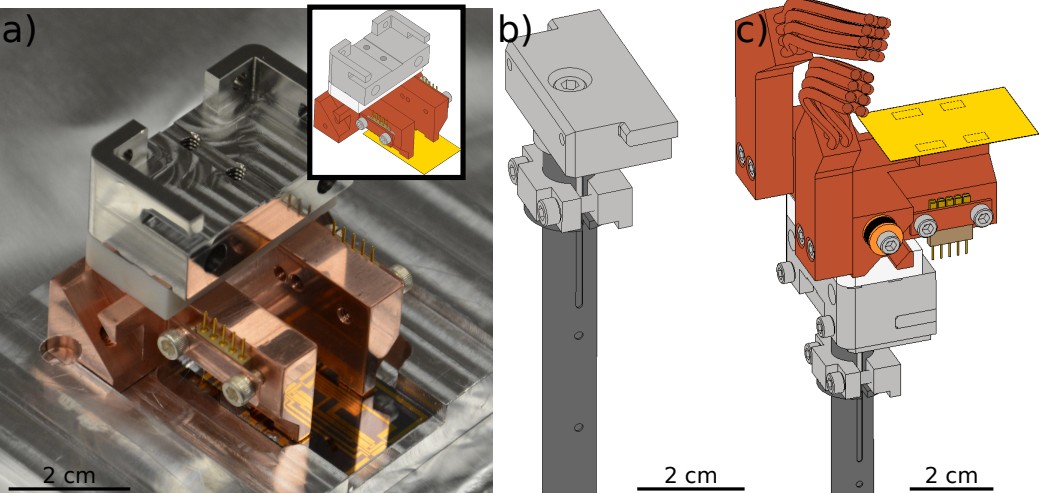

Figure 2: The sample mount. (a) The mount is shown resting upside down on an aluminum support scaffold outside vacuum. Inset: schematic of the sample mount viewed from a similar angle. (b) Diagram of the sample support tube in the vacuum chamber with sample removed. The sample mount slides onto the top of the block (shown in light gray) situated at the end of the support tube. It is secured in place by tabs at each side. (c) Diagram of the sample mount as it would appear within the vacuum chamber. Copper cooling braids attach to the sample mount with screws.

connects to the blank flange, allowing arbitrary positioning of the sample in both position and angle with respect to the atoms. Controlled motion with the hexapod enables scanning over distances as large as 5 mm with precision ∼400 nm and limited not by the hexapod repeatability (∼60 nm), but by vibrations. See Section 4.2.

The copper cooling braids do not touch the walls of the radiation shield. They are held away by a copper retainer of bent sheet metal bolted to the 4 K stage of the ULVI. This ensures that no thermal short between sample and shield occurs, but also constrains the motion of the braids during sample translation. This constraint strains the titanium support tube when the sample moves, bending it slightly and influencing sample positioning. Because of this effect, the sample typically moves in steps smaller than those of the moving platform of the hexapod positioner, and large motions are typically accompanied by slow drift of the sample position of order 20 $\mu$m over a few hours. These effects are compensated for by imaging the sample in the vertical direction to determine the actual sample position. The position can either be corrected by moving the hexapod or by relabeling the data afterwards with the actual position. Future improvements to the braid connection system will address this issue.

The sample mount is also designed to facilitate rapid and simple exchange of samples. A back panel on the radiation shield is easily removed, allowing the sample to be exchanged through a 6" ConFlat port on the vacuum chamber; the back panel is shaded red in Fig. 3c. The slotted design of the sample mount allows it to easily slide on and off of the support tube; it is secured in place with screws. Electrical connections from the sample to vacuum feedthroughs are made by two five-pin PEEK plugs on the sample mount that can be connected to cables in the vacuum chamber prior to sample insertion. Sample exchange can be completed within an hour, after which the vacuum chamber must be baked to achieve ultrahigh vacuum at room temperature. A new sample can be loaded and ready to measure after a few-day bake.

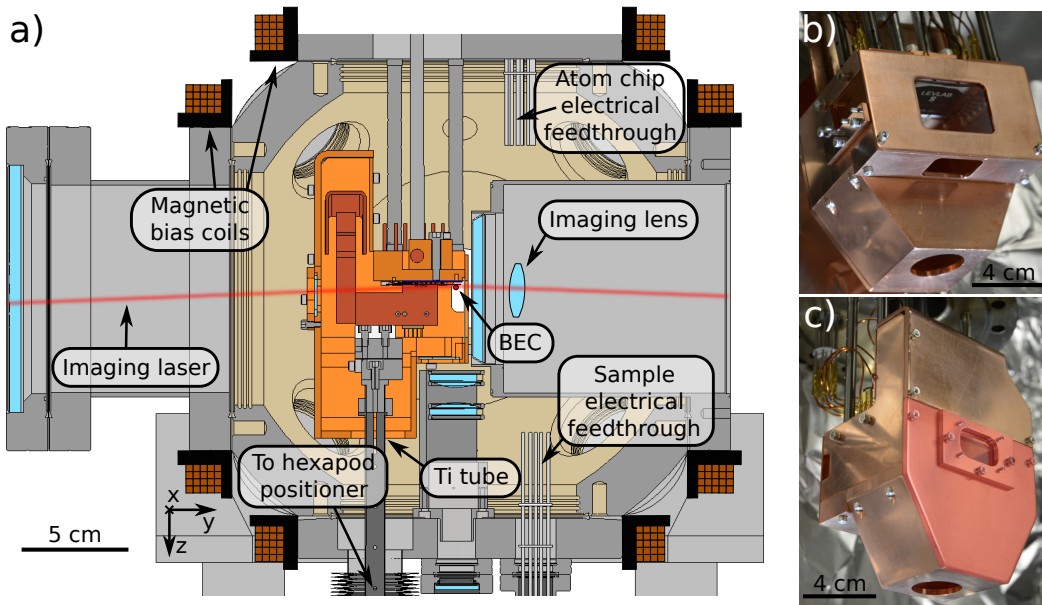

Figure 3: The radiation shield. (a) Cross section of the science chamber as viewed along the optical dipole trap axis. The BEC (red) is imaged using a 780 nm laser beam (transparent red) reflected off the sample at a shallow angle. (b, c) The assembled atom chip mount and radiation shield as viewed outside the apparatus from the (b) front and (c) back. The atom chip is visible through the front imaging window in (b). A portion of the shield, shaded red in (c), can be easily removed to load a new sample. The sample mount and the cryostat are not present in these images.

## 2.3 Radiation shield

Radiative heating is a major limiter of sample base temperature. The sample is exposed to blackbody radiation from both the vacuum chamber and the atom chip in the absence of shielding. Ideally, an enclosure should be built to shield the sample mount from the chamber walls and atom chip, but the close proximity of the sample to the atom chip prevents the insertion of a radiation shield between them. Instead, the atom chip and supporting structures are coupled to the radiation shield and cooled together; see Fig. 3. Note that this radiation shield is separate from that of the ULVI.

The wiring for the atom chip must be thin so as not to couple excessive heat into the cold radiation shield. Making the wires too thin, however, will raise the electrical resistance and cause significant ohmic heating under a load. As a reasonable compromise, we use 1 mm diameter wire (No. 18 AWG). The atom chip assembly must be kept cool to counteract ohmic heating even when the cryostat is off. This is because the SQCRAMscope is designed to run at room as well as cryogenic temperatures. To accomplish this, the atom chip assembly and sample radiation shield are completely decoupled from the cold finger and sample stage, as mentioned above. Cooling of the sample shield and atom chip is provided by a feedthrough to a liquid reservoir located above the atom chip. The lowest temperatures of the SQCRAMscope are reached when this reservoir is filled with liquid nitrogen. When operating above 9 K, however, it can instead be cooled by a closed-cycle chilled water loop. The atom chip and the surrounding radiation shield are supported by four hollow titanium tubes connected to a ConFlat blank flange on the vacuum chamber. The liquid feedthrough used for cooling the radiation shield and atom chip is supported by a flexible bellows to relieve strain caused by differential thermal contraction between it and the supporting titanium tubing.

This radiation shield is designed to preserve all optical access required by the SQCRAM-scope. A total of four cold sapphire windows are installed in the radiation shield: two for the imaging laser, one for the optical dipole trap (ODT), and one to allow vertical imaging of the sample. A hole, rather than a window, is placed on the production chamber side of the shield to allow the atoms to pass through when transported by the ODT. The imaging and ODT windows are held in place by silver-tipped set screws for thermal conductance, while the vertical imaging window rests unsecured in a groove. Using sapphire as the window substrate allows the windows to be made thin while maintaining high thermal conductance to the center of the window. This is of particular importance for the window used for high-resolution imaging of the atoms, which must fit in the small gap between the BEC and viewport. Finite-element simulations suggest that there is a thermal gradient of less than 0.2 K from the edge to the center of these sapphire windows, small enough to have no noticeable effect on the emitted blackbody radiation.

Unfortunately, some openings must remain in the sample radiation shield, in addition to the one on the side toward the production chamber. A hole on the bottom of the shield is required for the titanium tube that connects the sample mount to the bellows, enabling translational motion. Radiation through this hole could be blocked in the future by introducing a flexible shield such as a tight copper mesh around the tube. However, such a scheme would provide a thermal link between the two, and care would need to be taken to minimize thermal conduction. Finally, some small gaps remain where the sample radiation shield interfaces with the ULVI radiation shield.

# 3 Improvements to atom chip loading

The trapping scheme used in this new version of the SQCRAMscope is similar to that described previously [1]. However, some simplifications have been made to the atom chip trapping in the science chamber, allowing atoms to be trapped with fewer magnetic field wires. We now discuss these updated design elements of the atom chip and sample mount, as well as a brief description of the modified BEC preparation procedure.

## 3.1 Atom chip

The design of the atom chip is shown in Fig. 4. Electric current through three wires (central microwire, red; and two bias microwires, blue) runs parallel to the atomic gas along $\hat{x}$. The field from these wires creates a tight magnetic trap in the transverse ($y$-$z$) plane, while smaller currents in a pair of short wires oriented along $\hat{y}$ (green) provide longitudinal trapping along $\hat{x}$. A total of three pairs of longitudinal trapping wires are fabricated with spacings of 2.0 mm, 3.5 mm, and 5.0 mm. These three sets provide flexibility in the configuration of the longitudinal confinement field; larger spacing allows for a looser trap and longer gas. (Current is shown flowing through the middle pair of wires in Fig. 4.)

Large currents must run through the thin atom chip wires to form tight transverse atom chip traps due to the large, 300 $\mu$m distance to the atoms from the chip. This large distance is needed to allow room for the thin sample substrate chip to cantilever in between the atom chip and the BEC. The maximum sustainable current in the atom chip wires is determined by heat generation and dissipation during operation cycles [25]. The amount of heat generated is proportional to the resistance of a wire, and by extension, to the inverse cross-sectional area of the copper film. Heat is dissipated by conduction from the wires to the chilled water/liquid nitrogen reservoir, proportional to the width of the wires in contact with the silicon substrate.

The wire dimensions and positions are optimized to form a tight transverse magnetic trap at the typical operating distance. An 11.2 ± 0.1 $\mu$m-thick layer of copper on the atom chip

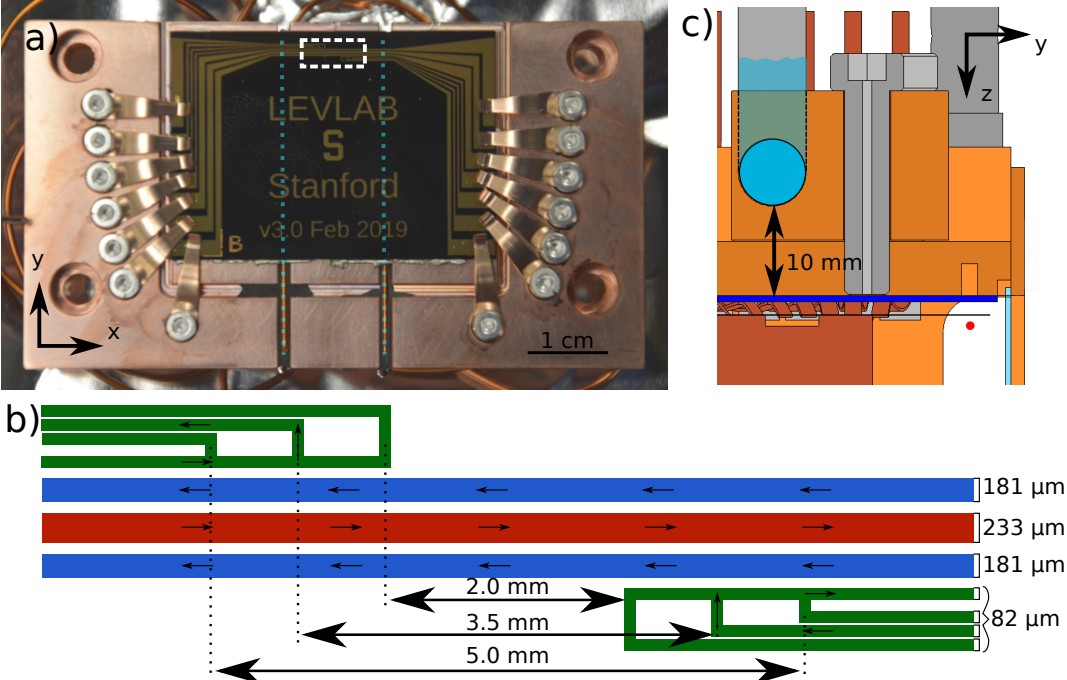

Figure 4: The atom chip. (a) Photograph of the mounted atom chip. The locations of additional longitudinal trapping wires located 1 mm behind the atom chip are indicated by dotted blue lines. (b) A sketch of the region outlined by the dashed box in panel (a) shows the wire portions that contribute most to the atom trapping. Current directions during measurement are as indicated. (c) Cross-sectional view showing the atom chip cooling apparatus. A reservoir (light blue) occupies a space 10 mm above the chip (dark blue) and is filled with liquid nitrogen or chilled water to cool both the atom chip and radiation shield. The distance between atom chip, sample chip, and atomic gas is exaggerated for clarity. The sample substrate is in black (a thin line, as seen on edge), the atom chip is in dark blue, the electrical contacts are in dark brown, the BEC position below the sample substrate chip is indicated by a red dot, and the viewport window for high-NA imaging is indicated by the light blue, thin vertical rectangle.

substrate is etched to form the microwires. Regarding wire width, a wider central microwire permits a larger breakdown current, but it also reduces the trap frequency at a fixed current. These two effects have to be properly balanced, along with similar constraints placed on the bias microwires and the requirement that they do not overlap. In our design, the central microwire is 233±4-$\mu$m wide, while the bias microwires are 181±4-$\mu$m wide, with 128±4-$\mu$m spacing between them. Uncertainties represent the precision with which we have measured these widths via an optical microscope.

Heat dissipation from the atom chip to the liquid cooling reservoir is optimized at all layers and interfaces. The thermally insulating Macor ceramic mount for the atom chip used previously [2] has been replaced by copper. This greatly improves thermal conduction to the cooling bath. With this modification, bulk thermal conductance from the chip to the cooling reservoir is increased from an estimated 0.18 W/K (using Macor) to 88 W/K (using Copper). This constitutes a factor of ∼500 improvement. Contact thermal resistance is now expected to limit the overall thermal conductance [25], since all intervening materials between the microwires and thermal bath—silicon, copper, and a thin native silicon oxide layer—have high thermal conductance. Indium is used at all joints to minimize this contact resistance: the atom

chip is soldered to the copper mounting block, and the copper reservoir for the cooling liquid is securely bolted to the back of the mounting block with a layer of indium in between. With chilled water cooling, currents in excess of 10 A, the maximum current allowed by the electrical feedthrough, may be flowed through the bias microwires or the central wire in steady state without detriment to the atom chip. This limit is well above the 5 A needed for trapping.

Electrical contacts are made to the atom chip using beryllium copper spring clips. The clips are screwed on one side to a beryllium copper pin, while the other end presses into the copper contact pads on the chip. The pins are secured to—and electrically insulated from—the copper atom chip mount with epoxy (Stycast 2850FT/catalyst 9). Wires connect to the back of the pins with standard push connectors, completing the connection to the chamber's electrical feedthroughs.

The science chamber pressure is $6 \times 10^{-11}$ Torr at room temperature (after a few-day bake) despite the great amount of intrachamber material. This enables BEC production and scanning probe measurement operation without cryopumping.

## 3.2 Ultracold atom production

The redesign of the atom chip and its mount allows us to simplify the atom trapping scheme of the SQCRAMscope. We now describe these changes.

An ultracold gas of $^{87}$Rb is prepared in the production chamber following the procedure described in Refs. [1,2]. The gas of 40 million atoms is then transferred into an ODT created by an 8 W, 1064 nm laser beam that is tightly focused by a lens on a translation stage. The atoms may be transported (optically tweezered) to approximately 3-mm below the atom chip trapping region by moving the translation stage. This distance allows the beam to clear the atom chip and sample without clipping. An acousto-optic deflector (AOD) then moves the focus of the ODT vertically, bringing the atoms approximately 0.9-mm below the atom chip. This motion improves the loading efficiency of the magnetic trap by allowing a tighter and deeper trap that better matches the optical potential. We measure a ∼30% improvement in BEC number using this AOD steering of the ODT.

Rather than this AOD steering, the previous version of the SQCRAMscope used a two-stage magnetic transfer involving the use of "macrowires" to capture the atoms from the optical tweezer. This was necessary because the previous atom chip could not support enough current in its microwires to create a sufficiently deep trap at the tweezer position. The field from macrowires under the chip was therefore needed to shuttle the atoms vertically from the tweezer to a position close enough for the atoms to be transferred to the microwire trap. This complicated the BEC preparation procedure. Moreover, the need for macrowires greatly reduced the efficacy of the cooling system of the atom chip assembly due to their high thermal conductance to the room-temperature chamber. The chip could not reach cryogenic temperatures.

We found that better heat sinking of the atom chip, as well as the addition of the AOD described above, rendered these wires obsolete. The improved cooling, described in Sec. 3.1, allows for much larger currents to be run in the microwires, enough to produce traps sufficiently deep to load the atoms directly from the ODT. The new system runs currents in the central and bias atom chip microwires in combination with the fields from out-of-vacuum magnetic bias field coils and a pair of perpendicular copper wires (for longitudinal trapping) to create the magnetic trap. Loading efficiency is roughly the same as achieved in the old system, yielding the same BEC populations as before ($4.1 \times 10^4$ atoms with AOD steering), and the new system is far simpler to optimize and run. The central and bias microwires are depicted in red and blue, respectively, in Fig. 4b. The perpendicular copper wires are highlighted in dotted blue in Fig. 4a, and the bias coils (mounted at the edges of the vacuum chamber) are shown in Fig. 3a.

Better heat sinking of the atom chip mount has also improved stability of the atom trap position. Previously, the distance from BEC to sample drifted over time scales longer than the experiment cycle [2]. This was caused by thermal expansion of the atom chip support structure as the atom chip generated heat. With improved cooling, thermal expansion is greatly reduced and such drift is no longer observed.

Once loaded into the magnetic atom chip trap, the bias microwire current is reversed to bring the atoms closer to the sample, to within $\sim$200 $\mu$m. Finally, the atoms are compressed and further evaporated (using RF fields) to below the quasi-1D quasicondensate degeneracy temperature [26, 27]. The gas is then ready for measurement. The BEC is maneuvered with the atom chip fields to within a micron of the sample before allowing its density to equilibrate in response to the inhomogeneous potential from the sample. Then, the gas is released for a 150 $\mu$s time-of-flight before imaging its density with a resonant laser reflected off the surface of the sample at a shallow angle. This light is collected in a high-numerical-aperture lens and recorded on a CCD camera, resulting in a FWHM spatial resolution of 2.2 $\mu$m, as described in Ref. [2]. We have also added high-resolution imaging along the vertical axis by installing an in-vacuum lens pair; see Ref. [4] for details regarding these lenses. This allows us to simultaneously image both the atoms and sample surface so as to spatially register the two.

The shortest duty cycle so far achieved is 16 s [2], limited mostly by the laser cooling and tweezer translation steps. Once the atoms are condensed into the quasi-1D BEC, the atoms can be moved to the measurement position underneath the sample and equilibrated to its potential within $\sim$30 ms. This is far shorter than the 550 ms quoted in Ref. [2]. The first 300 ms of that time was an unnecessary pause—at 10 $\mu$m, the sample is sufficiently far that its fields need never affect the BEC, and so no such pause is necessary. The latter 250-ms portion is reducible to 30 ms while still maintaining adiabaticity as the BEC is translated closer to the sample. The first 20 ms of this time allows the trap to move from $z = 10$ $\mu$m, roughly where the BEC is produced, to the final measurement distance of $\sim$1 $\mu$m below the sample without exciting sloshing motion; the transverse trap frequency along $\hat{z}$ exceeds a kHz. The BEC is then held in place for 10 ms before imaging to allow the atomic density to equilibrate in the potential from the sample.

## 4 Characterization

We now present the new cooling capabilities along with data demonstrating that the new cryostat design does not introduce detrimental vibrations.

### 4.1 Sample cooling

A typical cooldown of the cryostat is shown in Fig. 5, with temperature data plotted for the sensors marked in Fig. 1. When cooled by liquid nitrogen, the radiation shield reaches a base temperature of $80.0 \pm 0.8$ K in 4 hours. The second stage of the ULVI reaches a base temperature of $4.3 \pm 0.5$ K. The sample reaches a base temperature of $5.7 \pm 0.1$ K after 8 hours with the radiation shield cooled by liquid nitrogen, or $8.20 \pm 0.02$ K if the shield is cooled by chilled water. Uncertainties account for the precision of temperature sensors and temperature stability. With the radiation shield cold, the temperature sensor on the sample wafer mount reads the same temperature as the sensor on the copper block below to within 0.1 K, less than the measurement error (0.5 K) of the uncalibrated sensor on the copper block. This indicates sufficient thermal conduction to the end of the thin silicon chip. We note that when the radiation shield is warm, radiative heating creates a $\sim 0.5$ K gradient between the thermometers

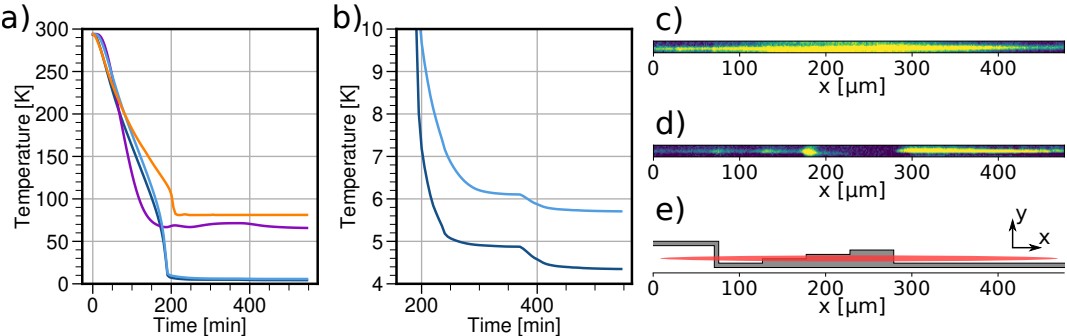

Figure 5: Typical cooling curve over (a) the full cool down period and (b) near base temperature. Temperatures are plotted for sensors labeled by triangles in Fig. 1: ULVI second stage (dark blue), sample mount (light blue), ULVI radiation shield (purple), and sample radiation shield (orange). The sensor on the sample substrate (not plotted) follows the sample mount (light blue) to within 1 K while cooling and less than 0.1 K when the temperature is stable. The liquid nitrogen-cooled radiation shield reaches a base temperature of 80 K in 4 hours, while the sample cools to 5.7 K over 8 hours. The downturns near 160 min and 200 min in the sample mount and radiation shield, respectively, are caused by a decrease in the specific heat of copper near 100 K [28]. The plateau and downturn in sample temperature near 370 min, as well as fluctuations in ULVI shield temperature from 150 min to 500 min, are caused by liquefaction of helium within the ULVI. (c,d) Absorption image of the atoms (yellow) trapped with the atom chip's magnetic fields 1-2 $\mu$m from a sample consisting of a patterned niobium superconductor. The Nb is cooled to (c) 9.3 K and (d) 9.0 K. These temperatures are above and below the superconducting transition temperature $T_c$, respectively. The fragmentation of the gas below Nb's $T_c$ may be ascribed to the distortion of the magnetic trap due to the Meissner repulsion of magnetic field through the nearby superconductor. (e) Sketch of the Nb film. Niobium is shown in gray, with the atomic gas location roughly indicated in red. Distortions of the gas density shown in panel (d) roughly correspond to the abrupt changes in the Nb wire width, as would be expected from the stray fields produced from these edges.

on the sample wafer and its mount due to finite thermal conductance from copper block to sample.

Figure 5 also shows a demonstration of our new ability to sense fields near the lower base temperature. A thermal gas of atoms was trapped near a patterned niobium superconducting thin film, shown schematically in Fig. 5e. The gas has a smooth cigar shape above the superconducting transition, as seen in Fig. 5c. However, the shape of the gas is severely distorted by Meissner screening of the magnetic trapping field when the Nb is cooled below its superconducting $T_c$; see Fig. 5d. We choose to employ a gas just warmer than the BEC critical temperature because we found, in this case, that the Meissner effect distortion induces a field variation greater than the BEC's dynamic range [2,5].

We estimate a radiative heat load of 33 mW on the 4 K stage, most of which comes through gaps in the shield that are for sample translation and optical transport of the rubidium atoms, as described in Sec. 2.3. At least 4 mW of heating is unavoidable due to the optical transport hole for inserting the atoms. Significant heating also comes from the sample support tube (160 mW). This heating could be reduced by cooling the sample support tube at an intermediate point with liquid nitrogen. Such a scheme is difficult due to physical constraints of the vacuum chamber, but may be achievable as a future upgrade.

Heat conducted through the long, thin electrical wiring to the sample mount is estimated to

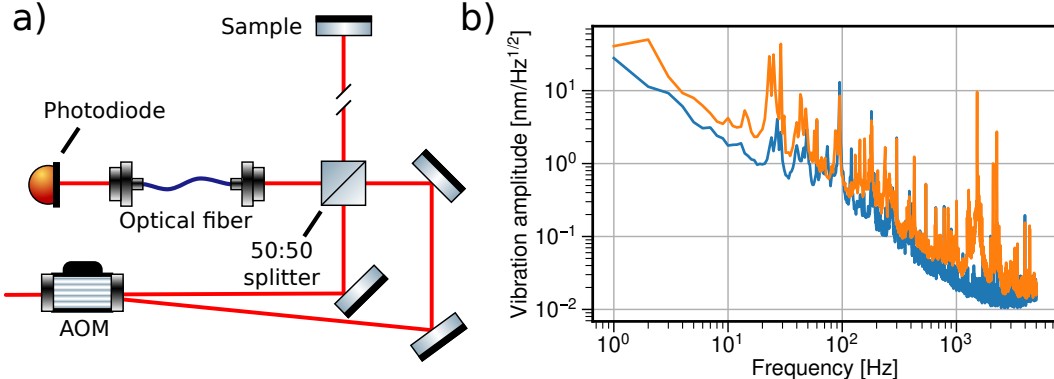

Figure 6: Vibration characterization. (a) Schematic of the interferometer setup used to measure vertical vibrations. A laser beam (red) is split by an acousto-optic modulator (AOM). The zero-order transmitted beam is reflected off the sample before being recombined with the first diffraction order on a beam splitter. The two beams are then coupled to an optical fiber and interfered on a photodiode. The resulting intensity signal contains a 70 MHz beat from the AOM frequency shift, as well as a slower signal caused by motion of the sample that changes the relative phase of the two beams. (b) The spectral density of vibrations is plotted with the cold head on at 6.1 K (orange) and off at room temperature (blue).

be 7 mW, and so is negligible in comparison to other sources. Increased thermal conductance from the 5.7 K sample to the 4.3 K ULVI region, e.g., through more robust copper braids linking the two, could further lower the sample temperature. However, thicker braids would further hamper sample motion.

## 4.2 Vibrations

Minimizing vibrations of the sample mount is critical to achieving micron-scale measurement resolution. Vibrations in the plane of the sample will blur field maps, while vibrations perpendicular to the plane of the sample, by changing the atom-sample distance, will impact both the effective spatial resolution and the measured magnitude of the sample field in a complex and sample-dependent manner. In both cases, vibrations should be kept well below the spatial resolution of the SQCRAMscope 2.2 $\mu$m [2] at the typical atom-sample distance of 1.0 $\mu$m. We use different techniques to measure vibrations in and out of the sample plane. We now describe these methods.

Vibrations in the plane of the sample are measured using an imaging system at normal incidence to the sample, reported previously [4] to have a resolution of about 3 $\mu$m. Images of the the sample are recorded at a rate of 7.5 Hz, limited by the frame rate of the camera, over a period of roughly 10 s. Relative displacements between these images are determined using the registration of recognizable features on the sample surface with an accuracy of 0.04 $\mu$m, as described in Appendix B. Sub-resolution and sub-pixel accuracy in image registration is possible when only a rigid 2D translation between images is present [29]. These measurements are ultimately limited by image signal–to–noise ratio. Using this technique, we measure a horizontal rms vibration amplitude of 0.4 $\mu$m. Typical long-term variation in rms vibration amplitude is 0.1 $\mu$m. The limited frame rate of the camera prevents us from observing the spectral weight of various vibrational frequencies in the plane of the sample.

Vibrations perpendicular to the surface of the sample are measured using a laser interferometer, as shown in Fig. 6a. To do so, a $\lambda = 780$-nm wavelength laser beam is first sent through an acousto-optic modulator (AOM). The first-order diffraction peak, shifted in frequency by

70 MHz relative to the incoming beam, is picked off by a mirror, sent through a beam splitter, and coupled into an optical fiber. This beam forms the reference arm of the interferometer. The zero-order transmitted beam of the AOM forms the sample arm of the interferometer. It travels through the same beam splitter as the reference arm, after which it is directed through a 1.33" viewport at the bottom of the UHV chamber. It then travels through the in-vacuum lens pair before retroreflecting off of a gold surface on the sample mount. This retroreflected beam is recombined with the reference arm of the interferometer at the beam splitter, after which both beams are coupled to an optical fiber and sent to a photodiode.

The signal from this photodiode contains a 70 MHz carrier modulation as well as modulation from the relative change in optical path length between the two arms. We assume a worst-case scenario that this path-length difference is entirely caused by vibrational motion of the sample mount rather than some other part of the interferometer or chamber. The 70 MHz modulation is removed from the photodiode signal using a commercially available quadrature amplitude demodulator, leaving the signal from the motion of the sample.

The phase of the collected signal exhibits a relative shift of $\phi = 4\pi\delta L/\lambda$ between the two arms of the interferometer, where $\delta L$ is the differential change in the relative path length between the two arms. The spectral density of vibrations is shown in Fig. 6b under the conditions in which the cryostat is on with the sample at base temperature (orange) and the cryostat is off (blue). As expected, the cryostat increases vibration amplitude at nearly all frequencies. Integrating over these vibration spectra in a frequency range from 1 Hz to 5 kHz results in an rms vertical vibration amplitude of $102\pm16$ nm with the cryostat running at base temperature and $42\pm16$ nm when off. These amplitudes are much smaller than the 2.2 $\mu$m spatial resolution of the SQCRAMscope and thus do not impact the SQCRAMscope's sensing capabilities.

We attribute the difference in magnitude between the vertical and horizontal vibration amplitudes to the dominant vibrational mode of the sample support tube. As a long tube, the primary vibrational mode is the flexing of the tube perpendicular to its axis. Because the support is oriented vertically, such bending will cause significantly more motion in the plane of the sample than in the vertical direction.

## 5  Conclusion

We have improved the cryogenic capabilities of the SQCRAMscope as well as simplified the sample loading and atom trapping procedures. A closed-cycle cryostat and radiation shield enable cooling to 5.7 K using a liquid-nitrogen-cooled radiation shield, or 8.2 K with the radiation shield cooled by water. The closed-cycle cryostat is far more cryogen-efficient than the previously used flow cryostat, and our design does not introduce vibrations detrimental to SQCRAMscope imaging resolution. The simplified atom trapping scheme involves fewer trapping wires than the previous design, which allows us to more efficiently cool the atom chip itself. Together, these improvements to the SQCRAMscope will enable the exploration of a wider variety of strongly correlated and topologically nontrivial quantum materials.

## Acknowledgments

We thank Stephen Edkins, Jenny Hu, and Josh Straquadine for contributions to the apparatus and Jun Wang for manuscript critique.

**Funding information**   We acknowledge funding support from the U.S. Department of Energy, Office of Science, Office of Basic Energy Sciences, under Award Number DE-SC0019174.

Fabrication of sample mount substrates and the atom chip were performed at the Stanford Nanofabrication Facility and the Stanford Nano Shared Facility, supported by the NSF under award no. ECCS-1542152.

## A    Atom chip fabrication

The new atom chip is fabricated on a float-zone silicon wafer with resistivity over 10 kΩ cm. This allows the trapping wires to be patterned directly on top of the wafer without the need for an electrically insulating layer in between. Such a layer would be thermally insulating, reducing heat dissipation from the trapping wires and, by extension, the maximum current capacity of the wires. We observe a resistance of $> 1$ kΩ between all non-connected wires, more than three orders–of–magnitude larger than the $< 1$ Ω resistance across the wires themselves. This indicates that the lack of a resistive layer poses no problem.

The microwires are mostly composed of a 11.2 $\mu$m layer of Cu evaporated onto the silicon wafer, with adhesion and capping layers above and below. Copper is used for its superior electrical and thermal conductivity. Wire patterning is done by standard photolithography techniques, followed by ion milling and wet etching.

## B    Image registration

We employ a masked image translation registration algorithm with sub-pixel precision [29,30]. A sample contains a region with a five-by-five grid of light-colored dots on a dark background designed for alignment purposes. A digital mask selects a small region of interest enclosing the grid for up-sampled cross-correlation computation with an up-sampling factor of 100. The algorithm achieves sub-pixel accuracy and precision.

We determine the precision of the image registration algorithm, and therefore our estimates of vibrations in the xy-plane, by reducing the region of interest to the upper and lower halves of the original region of interest, and compare the result of image registration on both halves. The precision, defined to be the standard deviation characterizing the variation in the registered image shift, is found to be 0.03 $\mu$m.

Similarly, the accuracy of the algorithm is determined by generating images with similar features and noise level as the experimental images, which are shifted by a known random distance and fed into the algorithm. It is found to be 0.04 $\mu$m.

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
