# Peer review of "A scanning quantum cryogenic atom microscope at 6 K"

_SciPost Physics, doi:SciPost Phys. 10, 060 (2021)_

## Round 1 · Referee Report · Anonymous (Referee 2) · 2021-1-26

Report

I am happy with the implemented changes and recommend the manuscript for publication.

---

## Round 1 · Author Response

Dear editor and referees,

Thank you for your helpful comments. We have edited the manuscript to include the requested revisions. All manuscript changes are highlighted in orange in the text. We now address referee comments:

Report 1

Requested changes:

1) ``If possible add a measurement result of the improved quantum gas microscope OR add a discussion section how the changes are expected to affect the quantum gas microscope measurements and why a direct demonstration is not possible at this point."

Thank you for this suggestion. We have added data showing the effect on the atoms of trapping near a Nb superconductor as we lower its temperature below its superconducting transition below 9 K. These data are in the new panels c-e of Figure 5 and discussed in the caption and in a new paragraph in section 4.1. This highlights the new capability of the SQCRAMscope to cool samples to lower temperatures.

``Additional minor points:"

2) ``Indicate confidence intervals or precision wherever possible (e.g. for measured temperatures or dimensions of wires on the chip)"

We added confidence intervals to atom chip microwire dimensions in Sec. 3.1. We also added temperature uncertainties to the base temperature results. The atom chip copper thickness in Sec. 3.1 has been updated to reflect a more precise measurement including a confidence interval.

3) ``Give more details on how the heat-sinking of the chip mount has been improved, since this seems to be an important change."

We added a sentence to Sec. 3.1 quantifying the improvement in thermal conductivity from switching from Macor to copper in the atom chip mount.

4) Page 10Loading efficiency is roughly the same..." How many atoms are loaded into the magnetic trap?"

A more important number is simply the total number of atoms in the BEC, which we now quote near this sentence. But to address the comment directly, the atom cloud is too large with respect to our camera field of view to quote a quantitative atom number at this intermediate stage. The core point is that it is qualitatively similar in size, yielding the important fact that the BEC population is the same.

5) ``Page 10 "...bring the atoms closer to the surface." How close are the atoms to the surface?"

We added `200~$\mu$m' to the text to quantify the distance of the atoms from the sample after bias microwire current reversal stage.

Report 2

Weaknesses

Weakness: ``It would have been great so see what the instrument is now capable to do. Showing how much improvement was achieved."

See similar comment in report 1 and our reply that we have now added data showing the trapping of the atoms near a Nb superconductor above and below its $T_c$.

Requested change:

"the references are very self centred"

We apologize for this oversight and have added the requested citations (Wildermuth2005, Aigner2008) to the introduction, in addition to a few others: see references 5, 19-22.

Additional manuscript changes

Corrected typo in Figure~5 caption: sensors in Fig.~1 were incorrectly labeled as circles instead of triangles.

Added additional acknowledgment for Stephen Edkins.

---

## Round 1 · List of Changes

List of changes:

Corrected typo in Figure 5 caption: sensors in Fig. 1 were incorrectly labeled as circles instead of triangles.

Added additional acknowledgment for Stephen Edkins.

Added data to Fig 5 showing the effect on the atoms of trapping near a Nb superconductor and text explaining this.

Added confidence intervals to atom chip microwire dimensions in Sec. 3.1

Updated atom chip copper thickness in Sec. 3.1 to reflect new, more precise measurement with confidence interval.

Clarified that the loading efficiency results in the same BEC population as before.

We also added temperature uncertainties to the base temperature results.

Added number clarifying 200~$\mu$m distance from atoms to sample after bias microwire current reversal in Sec. 3.2.

Added sentence quantifying the improvement in thermal conductivity from switching from Macor to copper in the atom chip mount in Sec. 3.1.

Added the requested citations (Wildermuth2005, Aigner2008) in the introduction, as well as a few others.

---

## Editorial Decision

published